# Pancreatitis Risk Associated with GLP-1 Receptor Agonists, Considered as a Single Class, in a Comorbidity-Free Subgroup of Type 2 Diabetes Patients in the United States: A Propensity Score-Matched Analysis

**DOI:** 10.3390/jcm14030944

**Published:** 2025-02-01

**Authors:** Mark Ayoub, Harleen Chela, Nisar Amin, Roberta Hunter, Javaria Anwar, Veysel Tahan, Ebubekir Daglilar

**Affiliations:** 1Department of Internal Medicine, Charleston Area Medical Center, West Virginia University, Charleston, WV 25304, USA; nisar.amin@vandaliahealth.org (N.A.);; 2Division of Gastroenterology and Hepatology, Charleston Area Medical Center, West Virginia University, Charleston, WV 25304, USA

**Keywords:** pancreatitis, diabetes, GLP-1 RA, weight loss, safety

## Abstract

**Introduction:** Glucagon-like peptide-1 receptor agonists (GLP-1 RAs) are commonly prescribed for the management of type 2 diabetes mellitus (T2DM). However, the potential connection between GLP-1 RAs and the risk of pancreatitis presents a complex and nuanced issue. Although these drugs are effective in improving blood sugar control and cardiovascular health, their association with pancreatitis remains an area of concern. Our study aims to evaluate the association between the use of GLP-1 RAs, considered as a single class, and the risk of pancreatitis in a comorbidity-free subgroup of patients with type 2 diabetes mellitus (T2DM) in the United States. **Methods:** Data were retrieved from the TriNetX research database using the US Collaborative Network, which included information from 61 healthcare organizations within the U.S. Patients diagnosed with T2DM were categorized into two cohorts: one consisting of the patients prescribed with GLP-1 RAs and the other comprising patients who did not receive GLP-1 RAs. Of this class of medications, the agents analyzed were dulaglutide, lixisenatide, exenatide, liraglutide, and semaglutide. Using a 1:1 propensity score matching (PSM) model, we matched patients of both cohorts based on baseline demographics, comorbidities (hypertensive disorders, ischemic heart disease, gallstones, annular pancreas, alcohol use disorders, hypertriglyceridemia, hypercalcemia, cystic fibrosis, and cannabis use), medications known to cause drug-related pancreatitis, and laboratory values. **Results:** Of 969,240 patients with T2DM, 9.7% (93,608) were on GLP-1 RA, and 90.3% (875,632) were not. After PSM, the sample included 81,872 patients in each cohort. The risk of pancreatitis between the two groups was not statistically different between the two cohorts at 6 months at (0.1% vs. 0.1%, *p* = 0.04), and remained without significant increase with time; at 1 year (0.1% vs. 0.2%, *p* = 0.02), 3 years (0.2% vs. 0.3%, *p* = 0.001), and 5 years (0.3% vs. 0.4%, *p* < 0.001). The lifetime risk of developing pancreatitis in patients on GLP-1 RA was lower (0.3% vs. 0.4%, *p* < 0.001). **Conclusions:** In our comorbidity-free U.S.-based population with T2DM, the use of GLP-1 RAs did not increase their risk of pancreatitis. Their use was associated with a lower lifetime risk of pancreatitis.

## 1. Introduction

Glucagon-like peptide-1 (GLP-1) agonists have become a cornerstone in the management of type 2 diabetes mellitus (T2DM), encompassing agents such as exenatide, liraglutide, and dulaglutide. These medications exert their glucose-lowering effects by emulating the action of endogenous GLP-1, stimulating insulin secretion, and inhibiting glucagon release. Due to their efficacy in enhancing glycemic control, promoting weight loss, and demonstrating cardiovascular benefits, GLP-1 RAs have witnessed a steady increase in prescription rates, contributing significantly to the treatment landscape for T2DM [1,2]. The American Diabetes Association (ADA) updated their guidelines, and they recommend the use of GLP-1 RAs as one of the first line therapies for T2DM [3].

Despite the therapeutic advantages offered by GLP-1 RAs, concerns have emerged regarding their potential association with an increased risk of pancreatitis [4,5,6]. Pancreatitis, marked by inflammation of the pancreas, is a serious condition with potentially severe consequences. The existing literature has seen several studies exploring this potential link, prompting a comprehensive examination of the risk–benefit profile of GLP-1 RAs in the context of pancreatitis [4,5,7,8].

Singh et al., in a study from 2013, showed through an administrative database study of U.S. adults with type 2 diabetes mellitus that treatment with the GLP-1-based therapies sitagliptin and exenatide was associated with the increased odds of hospitalization for acute pancreatitis [5]. However, another study by Storgaard et al. in 2017 found no evidence that treatment with GLP-1 RAs increases the risk of AP in patients with type 2 diabetes [4].

According to the U.S. Centers for Disease Control and Prevention (CDC), in 2021, there were 38.1 million adults above the age of 18 (14.7% of U.S. adults) with diabetes [9]. The percentage of adults with diabetes increases with age reaching 29.2% among those older than 65 years [9]. Some studies found that a majority of adult patients with T2DM have at least an additional comorbidity, and almost 40% of adult patients have at least three additional ones [10]. In one of those studies, 97.5% of 1.3 million patients had at least one comorbidity in addition to T2DM [11]. The strong association of other comorbidities with T2DM was the basis of our study structure with their exclusion.

Due to such conflicting data and the severe nature of acute pancreatitis, further research is warranted to investigate this potential link to allow for safer prescribing methods. The available studies in the literature focus on specific drugs within the GLP-1 RA class, without addressing the remainder of available drugs. Therefore, our study objective was to evaluate the association between the use of GLP-1 receptor agonists (GLP-1 RAs), considered as a single class, and the risk of pancreatitis in a comorbidity-free subgroup of patients with type 2 diabetes mellitus (T2DM) in the United States, using propensity score matching (PSM) to balance baseline characteristics.

## 2. Materials and Methods

### 2.1. Statistical Analysis

This study was approved by the Institutional Board Review Committee at Charleston Area Medical Center. Written informed consent from patients was waived due to the de-identified nature of the TriNetX clinical database. The TriNetX (Cambridge, MA, USA) database is a global federal research network that combines real-time data with electronic medical records. Data used in our study were retrieved from the TriNetX research database using the US Collaborative Network, which included information from 61 healthcare organizations within the U.S. Using the International Classification of Diseases 10th revision codes (ICD-10), adult patients aged ≥ 18 years with type 2 diabetes mellitus (T2DM) were identified. Included patients were divided into two cohorts that underwent propensity score matching (PSM) and were subsequently compared. We matched patients of both cohorts based on baseline demographics, comorbidities, medications known to cause drug-related pancreatitis, and laboratory values.

After performing PSM, outcome analysis was performed. Our outcome for this study was the risk of pancreatitis between the two cohorts, which was analyzed using Kaplan–Meier curves and log-rank tests. Risk ratios (RR) with 95% confidence intervals (CI) were calculated for our outcome. A *p*-value of <0.05 was considered statistically significant. All statistical analyses were conducted on the TriNetX platform on https://live.trinetx.com/ (accessed date 31 March 2024, 23:35:21 UTC).

### 2.2. Inclusion and Exclusion Criteria

Patients with T2DM were divided into two cohorts: the first cohort comprised patients who received GLP-1 RAs, and the second cohort comprised patients who did not receive GLP-1 RAs. The selected agents of the GLP-1 RA class that were used in our study were dulaglutide, lixisenatide, exenatide, liraglutide, and semaglutide. We excluded patients with type 1 diabetes mellitus, heart failure, ischemic heart diseases, hypertension, and chronic kidney disease. We compared the risk of pancreatitis between the two cohorts over time using a 1:1 PSM model using patients’ baseline characteristics as well as risk factors commonly associated with the development of pancreatitis. PSM components included demographics, comorbidities, medications used that are known to cause drug-related pancreatitis, and lab values. Demographics included age at index, race, gender, and ethnicity. Comorbidities included, but were not limited to, hypertensive disorders, ischemic heart disease, gallstones, annular pancreas, alcohol use disorders, hypertriglyceridemia, hypercalcemia, cystic fibrosis, and cannabis use. Medications included most medications and drug classes that were reported in the literature to be associated with pancreatitis. Lab values included hemoglobin A1C level, body mass index (BMI), triglyceride level, and calcium level. The full list of items used for PSM is shown in the Appendix A.

## 3. Results

### 3.1. Baseline Characterestics

A total of 969,240 patients with type 2 diabetes mellitus were identified. Of those, 93,608 (9.7%) patients were on a GLP-1 RA, and 875,632 (90.3%) patients were not on a GLP-1 RA. The mean age in the GLP-1 group was 47.3 with a standard deviation (SD) of 11.9. More than half the cohort comprised females at 62.6%. The mean body mass index (BMI) in the GLP-1 group was 35.6 kg/m^2^ with a SD of 6.9. In the GLP-1 group, alcohol use disorder was found in 1.4% and cholelithiasis was found in 2.1%. The mean triglyceride level in the GLP-1 group was 187 mg/dL with a SD of 175, and a calcium level of 9.4 mg/dL with a SD of 0.5. Of the GLP-1 group, 53.4% received metformin, 38.9% received glucocorticoids or mineralocorticoid, 32.5% received penicillins or a beta-lactam antibiotic, and 22.9% received non-steroidal anti-inflammatory analgesics (NSAIDs). A full comparison of the cohorts’ baseline demographics, comorbidities, medications, and lab values between the two cohorts before and after PSM is highlighted in the Appendix A.

### 3.2. Outcomes

Analysis of the cohorts’ baseline demographics, comorbidities, medications used, and lab values did not show any significant differences after PSM. We compared the rate of pancreatitis between the two cohorts after PSM over time. The rate of pancreatitis was similar in patients receiving GLP-1 RAs and those who did not receive GLP-1 RAs (0.1% vs. 0.1%, *p* = 0.035). Patients receiving GLP-1 RAs had a significantly lower rate of pancreatitis at one year (0.1% vs. 0.2%, *p* = 0.022), three years (0.2% vs. 0.3%, *p* < 0.001), and five years (0.3% vs. 0.4%, *p* < 0.001). The lifetime risk of pancreatitis was significantly lower in patients receiving GLP-1 RAs compared to those who did not receive GLP-1 RAs (0.3% vs. 0.4%, *p* < 0.001). The hazard ratio of the risk of pancreatitis over the study duration is shown in Table 1 and a summary of the results is highlighted in Table 2 and Figure 1.

## 4. Background and Discussion

Glucagon-like peptide-1 receptor agonists (GLP-1 RAs) is a class of common type 2 diabetes mellitus treatments with increasing uses. It is also used in obesity as a weight loss regimen. The American Diabetes Association recommends GLP-1 RA use in patients with type 2 diabetes mellitus who have a higher risk of atherosclerotic cardiovascular disease, heart failure, or chronic kidney disease [12]. Due to their favorable outcomes in terms of glycemic control and weight loss properties, GLP-1 RAs are becoming more commercially available. However, tolerability and cost remain a barrier to their prescription [13].

GLP-1 is secreted from L-cells in the distal ileum and colon in response to meals [14]. It binds to islet alpha and beta cells, and receptors in the central and peripheral nervous systems, heart, lung, kidney, and gastrointestinal tract [15]. Attachment to beta cells in the pancreas leads to insulin production in response to hyperglycemia [15]. Exogenous GLP-1 RA efficacy relies on the concentration of innate GLP-1 and the GLP-1 RA’s affinity to their receptors, as well as the number of bound receptors [15]. The more receptors that are bound, the more they are activated, and the better the glycemic response is. The sustained activation of GLP-1 receptors leads to increased insulin synthesis, beta cell proliferation and their resistance to apoptosis, and improved survival [9]. This further leads to the slowing of gastric emptying, a feeling of satiety, the inhibition of glucagon secretion, and the subsequent improvement of glycemic control [14,15,16,17].

The safety profile of GLP-1 RAs is being extensively studied, with evidence suggesting a generally favorable risk–benefit profile, though close monitoring is advised, particularly for patients with specific comorbidities [18]. As GLP-1 receptors are present in tissues throughout the body, including in the nervous system, gastrointestinal tract, cardiac tissue, thyroid, and retina, the potential adverse effects are likewise widespread [19]. The most reported side effects are self-limited gastrointestinal symptoms, including nausea, vomiting, and diarrhea, often associated with the initiation of treatment and increases in dosage [20]. Most reported patients who have developed an AKI had risk factors contributing to volume depletion, or comorbidities that increased their risk, or took other nephrotoxic medications in combination with GLP-1 RAs [18]. The protective effects of GLP-1 RAs likely confer more significant benefits than risks to the kidneys [21,22]; studies have demonstrated the efficacy of GLP-1 RAs in slowing the progression of diabetic kidney disease [22]. Initially, there was also a concern over the possible increased risk of thyroid cancer associated with GLP-1 RAs, leading the FDA to issue a black box warning for specific agents; however, population-based studies have yielded more mixed results and no causal link has yet been established [23,24]. Rare ocular side effects such as a higher risk of diabetic retinopathy, complications from diabetic retinopathy, and non-arteritic ischemic optic neuropathy have been documented in the literature; these complications may be due to rapid reductions in A1c rather than as a direct result of GLP-1 RAs [18,25].

### 4.1. Mechanism

Proglucagon is a protein in enteroendocrine cells, alpha cells of the pancreas, and in the brainstem. Proglucagon undergoes post-translational processing by a convertase enzyme to produce glucagon, Glicentin-related pancreatic polypeptide (GRPP), and the major proglucagon fragment (MPGF) [26]. This MPGF contains both glucagon-like peptide (GLP)-1 and GLP-2. Peripherally, the main source of GLP-1 is the enteroendocrine GLP-1-producing cells in the small intestine and colon [27]. Centrally, GLP-1 is also produced by the brain in a similar fashion to neuroendocrine cells [28].

GLP-1 secretion from enteroendocrine cells is triggered by carbohydrates, proteins, and fats [27,29,30,31]. Each of these act differently on the enteroendocrine cells and eventually lead to the extraction of GLP-1. This is crucial for its release as studies proved that IV administration of glucose, for instance, does not lead to a change in GLP-1 levels in the blood [32]. This means that these nutrients have to pass through the gut to trigger GLP-1 secretion.

GLP-1 is also secreted centrally from the brain by the release of preproglucagon neurons. They become activated after gastric distension or in a response to enzymes or hormones such as cholecystokinin and leptin [33,34,35]. Furthermore, the peripheral secretion of GLP-1 triggers its own central secretion [26]. Peripherally-released GLP-1 activates GLP-1 receptors on the vagus nerve, which in turn stimulates centrally-located preproglucagon neurons and the subsequent secretion of GLP-1 [36,37,38].

GLP-1 has a very short half-life with a maximum of 11 min [39,40,41]. Peripherally, GLP-1 is mainly metabolized by the liver and endothelial cells [42]. In order to be able to clinically contribute to diabetes management, a long-acting version of GLP-1 is needed to be synthesized. Exendin-4 was discovered in a venom of a lizard in southwestern USA and was found to have a 53% resemblance to the innate GLP-1 [43]. The synthetic version of exendin-4 is currently available commercially as Exenetide, which is slower to be metabolized with a half-life of 4 h [44]. Further biochemical adjustments led to the synthesis of Liraglutide, which has 97% similarity to the innate GLP-1 [44]. Further advancements and biochemical improvements led to the synthesis of the remainder of the commercially available GLP-1 RAs.

### 4.2. Comparison

Exenatide and lixisenatide are exendin-4 derivatives and are typically administered daily 60 min before meals. Liraglutide and oral semaglutide are modified human GLP-1 and are taken any time daily. Dulaglutide and subcutaneous semaglutide are administered once weekly [45].

From a metabolic standpoint, multiple clinical trials were conducted to compare these GLP-1 RAs [46]. One meta-analysis showed that once-weekly GLP-1 RA administration had better hemoglobin A1c (HbA1c) reduction when compared to twice daily exenatide. However, it was not better than once daily liraglutide injection [47]. Another meta-analysis showed that liraglutide had better HbA1c reduction than exenatide but was not significantly different when compared to dulaglutide. The LEAD-6 trial and the DURATION-6 trial further confirmed the previous statement and showed that liraglutide, when compared to exenatide, had better reduction in HbA1c and weight with fewer adverse effects [48,49].

The AWARD-1 trial showed that dulaglutide was superior to exenatide, while the AWARD-6 trial showed that it was not inferior to liraglutide [50]. The SUSTAIN-7 trial compared semaglutide to dulaglutide which showed the superior effect of semaglutide in glycemic control and weight loss, it also showed they have a similar safety profile [50].

From a cardio–renal standpoint, there were seven clinical trials done on seven GLP-1 RAs that showed that liraglutide, subcutaneous semaglutide, and dulaglutide all showed significant cardiovascular risk reduction [51]. The LEADER trial showed that liraglutide had a cardiovascular benefit in high-risk patients, while the SUSTAIN-6 trial showed a cardiovascular-associated mortality reduction with semaglutide use as well as albuminuria reduction [46]. The SUSTAIN and PIONEER trials showed that semaglutide has a significant risk reduction in major adverse cardiovascular events [52]. A list of the previously mentioned trials with the studied GLP-1 RA used in each and their overall benefit is shown in Table 3 and Table 4 [53].

### 4.3. Pancreatic Involvement and Available Data

Due to their mechanism of action via attachment to alpha and beta cells in the pancreas and their subsequent stimulation, the concern for the occurrence of pancreatitis was raised [59]. The Food and Drug Administration (FDA) issued a warning due to concerns about GLP-1 RAs and the increased risk of pancreatitis and pancreatic cancer [60]. This was further reinforced in their latest release in 2024 [61,62]. This was recently debunked by very recent studies that did not show any association between GLP-1 RA use and pancreatic cancer [63,64,65].

Dulaglutide carries a boxed warning that it may increase the risk of pancreatitis [66,67]. Lixisenatide, according to its manufacturing label, was associated with cases of pancreatitis, and an alternative antidiabetic regimen should be pursued [68,69]. There were also post-marketing reports of acute pancreatitis in patients taking exenatide which were submitted to the Food and Drug Administration (FDA) and were published in the literature [70,71,72,73]. Some reports showed that liraglutide at different doses is associated with an increase in lipase levels up to 30% [74,75,76]. This prompted the manufacturer’s labeling to recommend the prompt discontinuation of liraglutide if pancreatitis is suspected [77]. Similarly, some other case reports were published associating liraglutide with pancreatitis, which prompted their labeling to recommend against their use if pancreatitis is suspected [78,79,80,81].

Other conflicting studies debunked such associations [7,8,73,82,83]. A systematic review in 2014 of randomized and non-randomized trials by Ling Li et al. was unable to find convincing evidence that GLP-1 RAs increase the risk of pancreatitis [84]. The FDA and the European Medicines Agency (EMA) conducted a comprehensive evaluation of GLP-1 RAs due to the rising post-marketing reports of pancreatitis and pancreatic cancer [85]. The FDA evaluated more than 250 toxicology studies microscopically, which did not reveal any findings of pancreatitis [85]. Both the FDA and the EMA were not able to reach a final conclusion about such an association, however, they both agreed that the totality of the available data at that time was reassuring [85]. Another study by Meier and Nauck evaluated the pooled pancreatitis ratio in clinical trials ongoing at that time [86]. They saw a slightly elevated risk of pancreatitis with GLP-1 RA use; however, the number of incident cases were very small, and the statistical power was limited [86]. These findings aligned with our study outcome over time. The risk of pancreatitis was very similar between the cohort receiving GLP-1 RAs and those who did not receive GLP-1 RAs. Over the course of five years, the patients who received GLP-1 RAs had a statistically significant lower risk of pancreatitis than those who did not. The risk of pancreatitis in both cohorts was very low, which also confirmed the findings of the current literature [84,85]. This debunked the association of pancreatitis incidence with GLP-1 use. In fact, patients receiving GLP-1 RAs had a significantly lower rate of pancreatitis in their lifetime. Our findings should allow physicians to safely prescribe GLP-1, especially as their cardiovascular and renal benefits outweigh the very minimal risk of a pancreatitis incident. A list of some of the available GLP-1 RA studies with the reported pancreatic outcomes is shown in Table 5.

### 4.4. Strengths and Limitations of Our Study

Some of our study strengths included the use of a nationwide database and the inclusion of a large cohort of patients. Additionally, using a 1:1 PSM model allowed us to reduce the effects of confounding the baseline covariates by creating matched cohorts, in which the distribution of the measured baseline covariates was similar in treated and control participants. This method created very similar cohorts after PSM and allowed us to clarify a more precise association while minimizing confounders. Furthermore, the exclusion of patients with CKD and using BMI in the PSM excluded and accounted for other indications for the use of GLP-1 RAs and ensured that their use in our patient population was for glycemic control.

Our study did not come without limitations. One of our study limitations was the use of the GLP-1 RA class as a whole without segmentation into different agents. This limited us from studying the variations between each single agent and their safety profile. Further studies and clinical trials are needed to explore such outcomes. Furthermore, despite the inclusion of a large number of patients and using a 1:1 PSM model, the use of a U.S.-based database and the exclusion of patients with multiple comorbidities limited the generalizability of our findings. We also focused on the risk of pancreatitis with GLP-1 RA use and did not explore other possible associations, such as the risk of solid organ malignancies, which warrant more clinical studies to further clarify the GLP-1 RA safety profile.

## 5. Conclusions

In our U.S.-based population, the use of GLP-1 RAs in patients with T2DM did not seem to increase their risk of pancreatitis. In fact, it was associated with a lower rate of pancreatitis in these patients compared to those not being treated with a GLP-1 RA. Although this study focused on the risk of pancreatitis, the cardiovascular, renal, and metabolic benefits of GLP-1 RAs, which are well-documented in the literature, reinforce the relevance of this drug class and suggest that their continued use might be favorable. More studies and clinical trials are needed to explore the risk of pancreatitis by each individual available agent of the GLP-1 RA class.

## Figures and Tables

**Figure 1 jcm-14-00944-f001:**
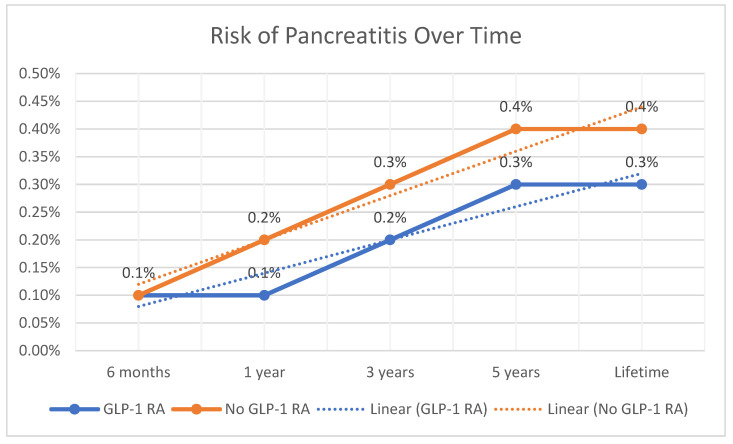
Linear graph of the risk of pancreatitis over time.

**Table 1 jcm-14-00944-t001:** Chronological hazard ratio of the risk of pancreatitis.

	Hazard Ratio	95% CI	*p*
6 Months	0.690	(0.503, 0.947)	0.461
1 Year	0.748	(0.584, 0.958)	0.175
3 Years	0.812	(0.671, 0.983)	0.506
5 Years	0.814	(0.681, 0.972)	0.565
Lifetime	0.849	(0.716, 1.005)	0.575

**Table 2 jcm-14-00944-t002:** Table with a summary of the outcomes.

	6 Months			1 Year			3 Years			5 Years			Lifetime		
	GLP-1 RA *n* = 82,333	No GLP-1 RA *n* = 82,333	*p*-Value	GLP-1 RA *n* = 82,333	No GLP-1 RA *n* = 82,333	*p*-Value	GLP-1 RA *n* = 82,333	No GLP-1 RA *n* = 82,333	*p*-Value	GLP-1 RA *n* = 82,333	No GLP-1 RA *n* = 82,333	*p*-Value	GLP-1 RA *n* = 82,333	No GLP-1 RA *n* = 82,333	*p*-Value
Risk of Pancreatitis	0.1%	0.1%	0.035	0.1%	0.2%	0.022	0.2%	0.3%	<0.001	0.3%	0.4%	<0.001	0.3%	0.4%	<0.001
Risk Difference (CI)	−0.000 (−0.001, −0.0001)		0.035	−0.000 (−0.001, −0.0001)		0.022	−0.001 (−0.001, −0.0001)		0.001	−0.001 (−0.002, −0.001)		0.0001	−0.001 (−0.002, −0.001)		0.0001
Odds Ratio (CI)	0.713 (0.519, 0.978)			0.749 (0.585, 0.959)			0.724 (0.599, 0.877)			0.676 (0.567, 0.807)			0.650 (0.549, 0.768)		

**Table 3 jcm-14-00944-t003:** Clinical trials and their studied GLP-1 RAs.

Trial	Studied GLP-1 RA
LEAD-6 [48]	Exenatide Liraglutide
DURATION-6 [54]	Exenatide Liraglutide
AWARD-1 [55]	Dulaglutide Exenatide
AWARD-6 [56]	Dulaglutide Liraglutide
SUSTAIN-6 [57]	Semaglutide
SUSTAIN-7 [50]	Semaglutide Dulaglutide
LEADER [58]	Liraglutide
PIONEER [52]	Semaglutide

**Table 4 jcm-14-00944-t004:** Trials’ comparative outcomes within the GLP-1 RA class.

GLP-1 RA	A1C	Weight
Exenatide	Low-Intermediate	Low
Lixisenatide	Low	Low
Liraglutide	High	High
Dulaglutide	High	Intermediate
Semaglutide	Highest	Highest

**Table 5 jcm-14-00944-t005:** GLP-1 RA studies with a pancreatic outcome.

Study	Effect on Pancreas
Romley 2012 [73]	No association between exenatide and acute pancreatitis
Lando 2012 [75]	GLP-1 RA is associated with increased lipase level in patients with T2DM
Elashoff 2011 [6]	Increased odds ratio of pancreatitis in patients taking exenatide
Singh 2013 [5]	Increased odds ratio of pancreatitis in patients taking exenatide
Meier 2014 [86]	Slightly elevated risk of pancreatitis with GLP-1 RA
Wenten 2012 [7]	No increased risk of pancreatitis with exenatide
Dore 2009 and 2011 [8,82]	No association between exenatide and acute pancreatitis
Dankner 2024 [64]	No increased risk of pancreatic cancer with GLP-1 RA in 7 years
Ayoub 2024 [63]	No increased risk of pancreatic cancer with GLP-1 RA in 7 years

## Data Availability

Available data are presented. Additional data are only available as permitted by a third party.

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
