# Peer review of "Pancreatitis Risk Associated with GLP-1 Receptor Agonists, Considered as a Single Class, in a Comorbidity-Free Subgroup of Type 2 Diabetes Patients in the United States: A Propensity Score-Matched Analysis"

_jcm, 2025, doi:10.3390/jcm14030944_

Round 1
Reviewer 1 Report
Comments and Suggestions for Authors
GLP-1 agonists have a potential risk of causing pancreatitis. The purpose of this research is to explore the relationship between the use of GLP-1 agonists and the risk of pancreatitis. I think this manuscript is interesting. At present, I have some questions about this manuscript.
1. Authors chose TriNetX database for data analysis. Why choose this database instead of other databases? What are the advantages of TriNetX database?
2. It is widely known that both GLP-1 agonists and DPP IV inhibitors have the potential risk of causing pancreatitis. The second group of patients did not receive GLP-1 agonist treatment. Did they receive DPP IV inhibitor treatment?
3. Why did the authors use 1:1 propensity-scored matching model (PSM) as a statistical method?
4. GLP-1 agonists not only have the potential risk of causing pancreatitis, but also cause thyroid tumors. Why did authors not simultaneously study the relationship between GLP-1 agonists and the risk of thyroid tumors?
5. Please clearly indicate the innovation of this research in the manuscript.
Author Response
GLP-1 agonists have a potential risk of causing pancreatitis. The purpose of this research is to explore the relationship between the use of GLP-1 agonists and the risk of pancreatitis. I think this manuscript is interesting. At present, I have some questions about this manuscript.
Q1. Authors chose TriNetX database for data analysis. Why choose this database instead of other databases? What are the advantages of TriNetX database?
A1. This is a very good question. TriNetX is a worldwide research network that encompasses multiple databases, some of which, are nationwide in the U.S.A. This allows us to capture a large number of patients across the nation that meet our inclusion criteria. It also relies on ICD codes and CPT codes which allows us standardization across multiple electronic medical records (EMRs).
Q2. It is widely known that both GLP-1 agonists and DPP IV inhibitors have the potential risk of causing pancreatitis. The second group of patients did not receive GLP-1 agonist treatment. Did they receive DPP IV inhibitor treatment?
A2. Thank you for your question. Both groups did not have DPP IV inhibitors in their inclusion criteria. However, to account for any possible incidence of initiating a DPP IV inhibitor, all DPP IV inhibitors were included in the PSM and used in the matching process to ensure comparability and to ensure that both groups are similar.
Q3. Why did the authors use 1:1 propensity-scored matching model (PSM) as a statistical method?
This is an excellent question. Using a 1:1 PSM model allows us to reduce the effects of confounding due to measured baseline covariates by creating a matched or weighted sample in which the distribution of measured baseline covariates is similar in treated and control participants. This method creates very similar cohorts after PSM and allows us to clarify a more precise association (i.e. the use of GLP-1 RA and the risk of pancreatitis) while minimizing any confounders.
Q4. GLP-1 agonists not only have the potential risk of causing pancreatitis, but also cause thyroid tumors. Why did authors not simultaneously study the relationship between GLP-1 agonists and the risk of thyroid tumors?
A4. This is an excellent idea! The association between GLP-1 RA and risk of thyroid tumor could be the idea of our next study (will need to change the entire study model and account for thyroid tumor risk factor and patient population). However, we wanted to focus on the risk of pancreatitis in this manuscript.
Q5. Please clearly indicate the innovation of this research in the manuscript.
A5. Thank you for pointing this out. The study sheds light on the risk of pancreatitis with GLP-1 RA use, which according to manufacturing labeling, is a contraindication. We aimed to highlight the low incidence of such risk, to continue the beneficial use of GLP-1 RAs. We added this to paragraph 1 under the introduction.
Reviewer 2 Report
Comments and Suggestions for Authors
Comments to authors
The authors carried out an interesting study on the association between GLP-1 and the risk of pancreatitis. Overall, it is easy to read, but it needs significant improvement. In particular, I cannot find the supplementary material cited by the authors, it would be interesting if they provided the hazard ratio, and finally, they should restructure the discussion to focus on the topic of their study.
Introduction
· Lines 49-52: Some citation is missing to justify this statement.
Methods
· Table 1: include in supplementary material.
· “Patients with type 2 diabetes mellitus were divided into two cohorts as in Figure 1;”: Remove the reference to Figure 1 in Methods. As you have included Figure 1 in Results, the reference to Figure 1 should be in Results.
· As the authors estimate the results with Kaplan-Meier, it would be interesting if they provided the hazard ratio and 95% confidence intervals, especially at 5 years.
Results
· “A 103 full comparison of cohorts’ baseline demographics, comorbidities, medications, and lab 104 values between the two cohorts before and after PSM is highlighted in the supplementary 105 material.”: I can't find the supplementary material.
· “The rate of pancreatitis was 110 similar in patients receiving GLP-1 agonist and those who are not receiving GLP-1 agonist 111 (0.1% vs 0.1%, p=0.04). Patients receiving GLP-1 agonists had a significantly lower rate of 112 pancreatitis at 1 year (0.1% vs 0.2%, p=0.02), 3 years (0.2% vs 0.3%, p<0.001), and 5 years 113 (0.3% vs 0.4%, p<0.001). The lifetime risk of pancreatitis was significantly lower in patients 114 receiving GLP-1 agonists compared to those who are not receiving GLP-1 agonists (0.3% 115 vs 0.4%, p<0.001). A summary of the results is highlighted in Table 2 & Figure 1.”: The authors use a statistical significance level of p < 0.05. Therefore, all these results would be statistically significant. On the other hand, the inclusion of one more decimal would be recommended.
Discussion
· Table 3 is not relevant, but Table 4 should not be included as it is completely outside the purpose of your study.
· Authors should discuss whether a difference in association was observed according to the type of GLP1 and, if this is not possible, whether previous authors have found a differential association according to the specific drug.
The authors should reconsider their discussion, as there is a lot of information that is not closely related to the objective of the study. I suggest the following structure:
· First paragraph: main findings of your study.
· Second and third paragraphs: interpretation and contextualisation of your results (explanation of the variability found, i.e. if there are studies showing a higher or lower risk of pancreatitis, why this might be the case; if your study found no harmful effect, mechanisms or reasons why there is no harmful effect; if this is consistent with other studies; etc.).
· Penultimate paragraph: clinical and research implications of your study.
· Last paragraph: limitations of the study. This section is very important, all studies have limitations and a good recognition of limitations is a strength of the study.
Conclusions
· “should encourage the 248 continued use of GLP-1 RAs in the treatment of T2DM specially with their proven cardi-249 ovascular, renal, metabolic, and mortality benefit.”: Authors should be more careful. The authors should suggest, not categorically state, that GLP-1s are recommended.
Author Response
Comments to authors
The authors carried out an interesting study on the association between GLP-1 and the risk of pancreatitis. Overall, it is easy to read, but it needs significant improvement. In particular, I cannot find the supplementary material cited by the authors, it would be interesting if they provided the hazard ratio, and finally, they should restructure the discussion to focus on the topic of their study.
Thank you for your feedback. We provided the hazard ratio to the results section under table 2.
Introduction
- Lines 49-52: Some citation is missing to justify this statement.
Thank you for pointing this out. We added the citations.
Methods
- Table 1: include in supplementary material.
We included the tables and graphs in the main study manuscript as per journal guidelines.
- “Patients with type 2 diabetes mellitus were divided into two cohorts as in Figure 1;”: Remove the reference to Figure 1 in Methods. As you have included Figure 1 in Results, the reference to Figure 1 should be in Results.
Thank you for pointing this out. We removed the reference.
- As the authors estimate the results with Kaplan-Meier, it would be interesting if they provided the hazard ratio and 95% confidence intervals, especially at 5 years.
This is an excellent remark. We included the Hazard Ratio and 95% confidence interval for the study timeline under table 2.
Results
- “A 103 full comparison of cohorts’ baseline demographics, comorbidities, medications, and lab 104 values between the two cohorts before and after PSM is highlighted in the supplementary 105 material.”: I can't find the supplementary material.
Thank you for pointing this out. We have included the supplementary material file to the revision.
- “The rate of pancreatitis was 110 similar in patients receiving GLP-1 agonist and those who are not receiving GLP-1 agonist 111 (0.1% vs 0.1%, p=0.04). Patients receiving GLP-1 agonists had a significantly lower rate of 112 pancreatitis at 1 year (0.1% vs 0.2%, p=0.02), 3 years (0.2% vs 0.3%, p<0.001), and 5 years 113 (0.3% vs 0.4%, p<0.001). The lifetime risk of pancreatitis was significantly lower in patients 114 receiving GLP-1 agonists compared to those who are not receiving GLP-1 agonists (0.3% 115 vs 0.4%, p<0.001). A summary of the results is highlighted in Table 2 & Figure 1.”: The authors use a statistical significance level of p < 0.05. Therefore, all these results would be statistically significant. On the other hand, the inclusion of one more decimal would be recommended.
Thank you for pointing this out we were using the approximation to closest 0.00 decimals. We have added the full value as per our statistical report and we also included it to the table.
Discussion
- Table 3 is not relevant, but Table 4 should not be included as it is completely outside the purpose of your study.
Thank you for highlighting this. We wanted to point out the clinical trials that included GLP1 RAs and which drug of study they used and their overall outcomes to point out very briefly the clear benefits that would outweigh the risks.
- Authors should discuss whether a difference in association was observed according to the type of GLP1 and, if this is not possible, whether previous authors have found a differential association according to the specific drug.
Thank you for highlighting this. We studied GLP1 RA as a class to show overall association. Unfortunately, we did not run a specific analysis for each drug under the class. This is potentially an area of study of ours in the near future. We pointed this out under section 4.4. Strengths and Weakness of our study.
The authors should reconsider their discussion, as there is a lot of information that is not closely related to the objective of the study. I suggest the following structure:
- First paragraph: main findings of your study.
- Second and third paragraphs: interpretation and contextualisation of your results (explanation of the variability found, i.e. if there are studies showing a higher or lower risk of pancreatitis, why this might be the case; if your study found no harmful effect, mechanisms or reasons why there is no harmful effect; if this is consistent with other studies; etc.).
- Penultimate paragraph: clinical and research implications of your study.
- Last paragraph: limitations of the study. This section is very important, all studies have limitations and a good recognition of limitations is a strength of the study.
Thank you for your very valuable and constructive feedback. We rephrased our study accordingly.
Conclusions
- “should encourage the 248 continued use of GLP-1 RAs in the treatment of T2DM specially with their proven cardi-249 ovascular, renal, metabolic, and mortality benefit.”: Authors should be more careful. The authors should suggest, not categorically state, that GLP-1s are recommended.
Thank you for helping us phrase the conclusion. We followed your recommendation.
Reviewer 3 Report
Comments and Suggestions for Authors
Dear Authors,
Respectfully, I have shared my comments and suggestions, which I hope will contribute to enhancing the clarity, coherence, and scientific rigor of your manuscript. Below, I outline key areas for improvement and propose practical recommendations to address them.
Title
Comment. The current title is too generic and does not reflect the scope of the study, which may lead to misinterpretations. I suggest revising the title to highlight specific aspects, such as: "Pancreatitis Risk Associated with GLP-1 Receptor Agonists, Considered as a Single Class, in a Comorbidity-Free Subgroup of Type 2 Diabetes Patients in the United States: A Propensity Score-Matched Analysis."
Introduction
Comment. Include data on the epidemiology of type 2 diabetes (T2DM) in the United States, highlighting the relevance of the excluded comorbidities.
Comment. The introduction does not address the pharmacological differences between short-acting and long-acting GLP-1 RAs, nor does it discuss how these differences may (or may not) impact pancreatic safety.
Objective
Comment. The current objective does not explicitly address the significant exclusions made, such as the exclusion of patients with common comorbidities. Additionally, it uses the term "GLP-1" without clarifying that the medications were analyzed as a single class, with no differentiation between short-acting and long-acting agonists. I suggest a reformulation, such as: "This study aims to evaluate the association between the use of GLP-1 receptor agonists (GLP-1 RA), considered as a single class, and the risk of pancreatitis in a comorbidity-free subgroup of patients with type 2 diabetes mellitus (T2DM) in the United States, using Propensity Score Matching (PSM) to balance baseline characteristics."
Methodology
Comment. Provide clear details on the Propensity Score Matching (PSM) method, specifying whether exact matching, nearest neighbor, or another technique was used.
Comment. Include visual and numerical evidence of group balance, such as love plots (standardized mean differences before and after matching) or balance tables. The absence of this information makes it difficult to assess the robustness of the model and compromises the replicability of the study. I suggest that this evidence be included as supplementary material (this file was not found).
Comment. Explain how missing data were handled.
Results
Table 2: The authors should include confidence intervals (CIs) for all pancreatitis rates and absolute differences between the groups. This data could be presented in the main table.
Comment. While statistically significant, the absolute differences in pancreatitis risk (0.1% to 0.2%) are small and raise questions about their clinical relevance. The absence of confidence intervals (CIs) further complicates the assessment of whether these differences are both statistically and clinically meaningful.
Background and Discussion
Comment. I recommend including a section dedicated to "Study Limitations".
Comment. The exclusion criteria used in the study were important to isolate the effect of GLP-1 RA use on pancreatitis risk, reducing the impact of confounding variables. However, PSM only balances observed variables and does not eliminate biases from unobserved factors, such as smoking, dietary habits, or genetic predispositions, which may significantly impact the results.
Comment. Unlike PSM, randomized clinical trials provide a more robust control against biases by randomly distributing both known and unknown variables across groups. Discuss how the findings might differ in a randomized clinical trial, which could provide more definitive evidence by eliminating confounding biases.
Comment. The exclusion of patients with common comorbidities such as hypertension and heart failure limits the generalizability of the findings. These patients represent the majority of the T2DM population in clinical practice. Therefore, these exclusions restrict the ability to generalize the findings to real-world populations, which should be explicitly acknowledged and discussed.
Comment. Dividing patients into two groups allows for a clear comparison between GLP-1 RA users and non-users. However, the study treats GLP-1 RAs as a single class, without exploring potential variations between different types of medications, such as short-acting (exenatide) and long-acting (dulaglutide, semaglutide). This approach simplifies the analysis but may obscure important or non-important differences related to individual drug characteristics.
Conclusion
Comment. Reassess the tone of the conclusion to avoid overly optimistic generalizations, emphasizing that the results should be interpreted with caution.
Comment. Explicitly acknowledge that the study did not segment the results by GLP-1 RA type, treating the medications as a single class, and that the exclusion of populations with multiple comorbidities limits the generalizability of the results. Suggest future studies that include these populations.
Comment. Include a brief mention in the conclusion that the study is based on U.S. data to provide proper context and ensure clarity regarding the population's geographic scope.
Comment. Emphasize that the study provides preliminary evidence, but that randomized clinical trials are needed to validate the findings and establish causality.
Kind Regards.
Author Response
Respectfully, I have shared my comments and suggestions, which I hope will contribute to enhancing the clarity, coherence, and scientific rigor of your manuscript. Below, I outline key areas for improvement and propose practical recommendations to address them.
Title
Comment. The current title is too generic and does not reflect the scope of the study, which may lead to misinterpretations. I suggest revising the title to highlight specific aspects, such as: "Pancreatitis Risk Associated with GLP-1 Receptor Agonists, Considered as a Single Class, in a Comorbidity-Free Subgroup of Type 2 Diabetes Patients in the United States: A Propensity Score-Matched Analysis."
Thank you very much for this constructive feedback! Your suggested title better represents our study structure. We changed our title to your suggested one.
Introduction
Comment. Include data on the epidemiology of type 2 diabetes (T2DM) in the United States, highlighting the relevance of the excluded comorbidities.
Thank you for pointing this out. We added a paragraph in the introduction.
Comment. The introduction does not address the pharmacological differences between short-acting and long-acting GLP-1 RAs, nor does it discuss how these differences may (or may not) impact pancreatic safety.
Thank you. We rephrased our introduction to reflect our study of GLP1 RA as a class.
Objective
Comment. The current objective does not explicitly address the significant exclusions made, such as the exclusion of patients with common comorbidities. Additionally, it uses the term "GLP-1" without clarifying that the medications were analyzed as a single class, with no differentiation between short-acting and long-acting agonists. I suggest a reformulation, such as: "This study aims to evaluate the association between the use of GLP-1 receptor agonists (GLP-1 RA), considered as a single class, and the risk of pancreatitis in a comorbidity-free subgroup of patients with type 2 diabetes mellitus (T2DM) in the United States, using Propensity Score Matching (PSM) to balance baseline characteristics."
Thank you so much for this constructive feedback! We have rephrased our introduction objective as you recommended.
Methodology
Comment. Provide clear details on the Propensity Score Matching (PSM) method, specifying whether exact matching, nearest neighbor, or another technique was used.
Thank you for pointing this out. We added the supplementary material of the PSM outcomes.
Comment. Include visual and numerical evidence of group balance, such as love plots (standardized mean differences before and after matching) or balance tables. The absence of this information makes it difficult to assess the robustness of the model and compromises the replicability of the study. I suggest that this evidence be included as supplementary material (this file was not found).
Thank you for pointing this out. We uploaded the supplementary file as recommended in the revision.
Comment. Explain how missing data were handled.
Thank you for your comment. We included all study data available.
Results
Table 2: The authors should include confidence intervals (CIs) for all pancreatitis rates and absolute differences between the groups. This data could be presented in the main table.
Thank you for the feedback. We included the risk difference, odds ratio, and the confidence interval for each in the main table.
Comment. While statistically significant, the absolute differences in pancreatitis risk (0.1% to 0.2%) are small and raise questions about their clinical relevance. The absence of confidence intervals (CIs) further complicates the assessment of whether these differences are both statistically and clinically meaningful.
Thank you for the feedback. We added the recommendations to the table.
Background and Discussion
Comment. I recommend including a section dedicated to "Study Limitations".
Comment. The exclusion criteria used in the study were important to isolate the effect of GLP-1 RA use on pancreatitis risk, reducing the impact of confounding variables. However, PSM only balances observed variables and does not eliminate biases from unobserved factors, such as smoking, dietary habits, or genetic predispositions, which may significantly impact the results.
Thank you, we added a limitations section.
Comment. Unlike PSM, randomized clinical trials provide a more robust control against biases by randomly distributing both known and unknown variables across groups. Discuss how the findings might differ in a randomized clinical trial, which could provide more definitive evidence by eliminating confounding biases.
Thank you, we added a limitations section.
Comment. The exclusion of patients with common comorbidities such as hypertension and heart failure limits the generalizability of the findings. These patients represent the majority of the T2DM population in clinical practice. Therefore, these exclusions restrict the ability to generalize the findings to real-world populations, which should be explicitly acknowledged and discussed.
Thanks for the feedback. We added it to our limitation section.
Comment. Dividing patients into two groups allows for a clear comparison between GLP-1 RA users and non-users. However, the study treats GLP-1 RAs as a single class, without exploring potential variations between different types of medications, such as short-acting (exenatide) and long-acting (dulaglutide, semaglutide). This approach simplifies the analysis but may obscure important or non-important differences related to individual drug characteristics.
This is a very good point. We highlighted this in the limitation section since our study focuses on the class of GLP-1 RA not individual regimens.
Conclusion
Comment. Reassess the tone of the conclusion to avoid overly optimistic generalizations, emphasizing that the results should be interpreted with caution.
Thank you for your feedback. We rephrased the conclusion.
Comment. Explicitly acknowledge that the study did not segment the results by GLP-1 RA type, treating the medications as a single class, and that the exclusion of populations with multiple comorbidities limits the generalizability of the results. Suggest future studies that include these populations.
Thank you, we added this to the conclusion.
Comment. Include a brief mention in the conclusion that the study is based on U.S. data to provide proper context and ensure clarity regarding the population's geographic scope.
Thank you, we added the clarification to the conclusion.
Comment. Emphasize that the study provides preliminary evidence, but that randomized clinical trials are needed to validate the findings and establish causality.
Thank you, we clarified in the conclusion as per your recommendations.
Kind Regards.
Thank you so much for your insightful and valuable feedback. This allows us to enhance our manuscript to better represent our study findings.
Round 2
Reviewer 2 Report
Comments and Suggestions for Authors
Comments to authors
I continue to suggest the convenience of sending certain tables in the supplementary material.
Author Response
I continue to suggest the convenience of sending certain tables in the supplementary material.
Thank you again for your feedback, we followed your recommendation and moved the previously mentioned table to the supplementary materials.
Reviewer 3 Report
Comments and Suggestions for Authors
Dear Authors,
I congratulate you on the excellent work in revising the manuscript. The supplementary material demonstrated that the analyses were conducted correctly and that the PSM model was effective in balancing baseline characteristics between the groups. The balance achieved is evident in the p values and the propensity score density after matching.
Below are some final suggestions.
Abstract
Comment: The objective does not highlight that the GLP-1 RAs were analyzed as a single class or that the studied population is comorbidity-free.
Comment: The conclusion omits the limitation of generalizing the study due to the exclusion of comorbidities and the focus on a U.S.-based population.
Conclusion
Comment: It is important to emphasize that the findings are limited to a comorbidity-free population.
Comment: Cardiovascular, renal, and metabolic benefits are mentioned without clarifying that these points are based on previous studies, not on the data from the current study. Suggestion: "Although this study focused on the risk of pancreatitis, the cardiovascular, renal, and metabolic benefits of GLP-1 RAs, which are well-documented in the literature, reinforce the relevance of this drug class."
Kind regards.
Author Response
Abstract
Comment: The objective does not highlight that the GLP-1 RAs were analyzed as a single class or that the studied population is comorbidity-free.
Thanks for the comment. We edited the abstract to reflect those recommendations.
Comment: The conclusion omits the limitation of generalizing the study due to the exclusion of comorbidities and the focus on a U.S.-based population.
Thank you for pointing this out. We followed your recommendations and edited the abstract to r
Conclusion
Comment: It is important to emphasize that the findings are limited to a comorbidity-free population.
We rephrased the conclusion to point this out.
Comment: Cardiovascular, renal, and metabolic benefits are mentioned without clarifying that these points are based on previous studies, not on the data from the current study. Suggestion: "Although this study focused on the risk of pancreatitis, the cardiovascular, renal, and metabolic benefits of GLP-1 RAs, which are well-documented in the literature, reinforce the relevance of this drug class."
Thank you so much for providing this insightful comment. We rephrased our conclusion to include your suggestion.
Kind regards.